# Early mortality in tuberculosis patients initially lost to follow up following diagnosis in provincial hospitals and primary health care facilities in Western Cape, South Africa

**Muhammad Osman**[1]*, **Sue-Ann Meehan**[1], **Arne von Delft**[2,3], **Karen Du Preez**[1], **Rory Dunbar**[1], **Florian M. Marx**[1,4], **Andrew Boulle**[2,3], **Alex Welte**[4], **Pren Naidoo**[1], **Anneke C. Hesseling**[1]

**1** Desmond Tutu TB Centre, Department of Paediatrics and Child Health, Faculty of Medicine and Health Sciences, Stellenbosch University, Cape Town, South Africa, **2** Centre for Infectious Disease Epidemiology and Research, School of Public Health and Family Medicine, Faculty of Health Sciences, University of Cape Town, Cape Town, South Africa, **3** Department of Health, Health Impact Assessment Directorate, Western Cape Government, Cape Town, South Africa, **4** DSI-NRF South African Centre of Excellence in Epidemiological Modelling and Analysis (SACEMA), Stellenbosch University, Stellenbosch, South Africa

* mosman@sun.ac.za

**Data Availability Statement:** Data cannot be shared publicly because of confidentiality accessing routine health service data. Data are

## Abstract

In South Africa, low tuberculosis (TB) treatment coverage and high TB case fatality remain important challenges. Following TB diagnosis, patients must link with a primary health care (PHC) facility for initiation or continuation of antituberculosis treatment and TB registration. We aimed to evaluate mortality among TB patients who did not link to a TB treatment facility for TB treatment within 30 days of their TB diagnosis, i.e. who were "initial loss to follow-up (ILTFU)" in Cape Town, South Africa. We prospectively included all patients with a routine laboratory or clinical diagnosis of TB made at PHC or hospital level in Khayelitsha and Tygerberg sub-districts in Cape Town, using routine TB data from an integrated provincial health data centre between October 2018 and March 2020. Overall, 74% (10,208/13,736) of TB patients were diagnosed at PHC facilities and ILTFU was 20.0% (2,742/13,736). Of ILTFU patients, 17.1% (468/2,742) died, with 69.7% (326/468) of deaths occurring within 30 days of diagnosis. Most ILTFU deaths (85.5%; 400/468) occurred in patients diagnosed in hospital. Multivariable logistic regression identified increasing age, HIV positive status, and hospital-based TB diagnosis (higher in the absence of TB treatment initiation and being ILTFU) as predictors of mortality. Although hospitals account for a modest proportion of diagnosed TB patients they have high TB-associated mortality. A hospital-based TB diagnosis is a critical opportunity to identify those at high risk of early and overall mortality. Interventions to diagnose TB before hospital admission, improve linkage to TB treatment following diagnosis, and reduce mortality in hospital-diagnosed TB patients should be prioritised.

available from the Western Cape Government
(https://nhrd.health.gov.za/) for researchers with
an approved protocol who meet the criteria for
access to confidential data (Email: Phdc.
Pgwc@westerncape.gov.za or Dr Melvin Moodley,
Director: Western Cape Government: Health
Impact Assessment: melvin.
moodley@westerncape.gov.za). The study protocol
may be accessed from the corresponding author.

**Funding:** This research and publication was
supported by the Bill and Melinda Gates
Foundation (BMGF), Investment ID: OPP1191594.
The contents are the responsibility of the authors
and do not necessarily reflect the views of the
BMGF. The work reported herein was made
possible through funding by the South African
Medical Research Council (SA MRC) through its
Division of Research Capacity Development under
the Bongani Mayosi National Health Scholars
Program from funding received from the Public
Health Enhancement Fund/South African National
Department of Health to MO. The contents of any
Publications from any studies during this Degree
are solely the responsibility of the authors and do
not necessarily represent the official views of the
SA MRC or the funders. ACH is financially
supported by the South African National Research
Foundation through a South African Research
Chairs Initiative (SARChI). FMM and AW are
supported financially by a grant from the South
African National Research Foundation through its
funding of the Centre of Excellence in
Epidemiological Modelling and Analysis. The
financial assistance of the NRF towards this
research is hereby acknowledged. Opinions
expressed, and conclusions arrived at, are those of
the authors and are not necessarily to be attributed
to the NRF. KDP is supported by the Fogarty
International Center of the National Institutes of
Health under Award Number K43TW011006. The
content is solely the responsibility of the authors
and does not necessarily represent the official
views of the National Institutes of Health. AB was
further supported by US National Institutes for
Health (grant numbers R01 HD080465 and U01
AI069924), the Bill and Melinda Gates Foundation
(grant numbers 1164272 and 119327), and the
Wellcome Trust (grant number 203135/Z/16/Z).
The funders had no role in study design, data
collection and analysis, decision to publish, or
preparation of the manuscript.

**Competing interests:** The authors have declared
that no competing interests exist.

## Introduction

In 2019, 10 million people developed tuberculosis (TB) and 1.4 million people (14%) with TB died [1]. South Africa is one of 8 countries which jointly carry two-thirds of the global TB burden [1]. An estimated 360,000 people developed active TB; the HIV prevalence was 58% in reported TB patients; and the estimated TB case fatality ratio (CFR) was 17% in 2019 [1]. In 2019, TB was the leading cause of death from a single infectious agent globally [1] and in South Africa TB continues to be the leading underlying cause of natural death [2].

Following testing for TB, further critical steps in the TB care cascade include the patient's receipt of their test results, initiation of treatment and linkage to TB care for TB treatment registration, and the continued management of TB. Gaps in TB treatment coverage have highlighted the burden of undiagnosed and untreated TB, with only 71% of incident TB cases being registered and reported, globally [1]. In a systematic review and meta-analysis of data from high burden TB settings in Africa, pre-treatment loss to follow up, defined predominantly as the gap between diagnosed TB patients and those recorded in TB treatment registers, was 18% [3]. Following diagnosis, TB patients may die or be lost before linking to TB care with or without having initiated TB treatment. These patients may remain unreported or return to health services for repeated testing and may eventually be included in treatment cohorts. We defined "initial loss to follow-up (ILTFU)" as TB patients who did not link to a TB treatment facility (primary health care (PHC) facility or TB hospital) for TB treatment within 30 days of their TB diagnosis having been made.

In 2019, South Africa registered 209,500 new patients with TB, an estimated treatment coverage of only 58%, with more than 150,000 people with TB left undiagnosed, or ILTFU (i.e. diagnosed but untreated and not registered in a TB treatment register) [1]. We aimed to estimate the burden of mortality amongst TB patients who were ILTFU. We hypothesised that mortality would be high among ILTFU TB patients and would differ depending on where the TB diagnosis had been made—at the hospital or PHC facility.

## Methods

### Study design

We prospectively evaluated mortality in adults and children routinely diagnosed with TB between 1 October 2018 and 31 March 2020 in two sub-districts in Cape Town, Western Cape Province, South Africa. We excluded all adults and children with confirmed drug-resistant TB. This analysis was nested in an operational research study aimed at reducing ILTFU using health system strengthening initiatives in these two large sub-districts, Khayelitsha and Tygerberg. Using integrated electronic reports, we identified all TB patients diagnosed in hospital and at PHC facilities and activated a system of linking patients to PHC facilities using a cascade of short message services, telephone calls, and community-based tracing of patients. We prospectively evaluated linkage to care at 30 days after TB diagnosis by evaluating attendance for TB treatment at any PHC facility in the province. We censored all patients at 7 months after TB diagnosis.

### Study setting

South Africa has a district health system with a focus on PHC for TB and HIV services. TB investigations, diagnosis, and treatment initiation may take place at any level of care. TB treatment registers are maintained at designated TB treatment sites and in Cape Town, this includes PHC facilities, and 2 designated TB hospitals. TB patients who are diagnosed at non-TB-designated hospitals may be initiated on TB treatment but typically receive 1–2 weeks of

medication and must be linked to a TB treatment site for continuation of their TB treatment and registration.

## Provincial health data centre

In the Western Cape Province, the Department of Health uses a provincial health data centre (PHDC) which harmonises electronic patient data from all public sector services in the province and produces a single patient record [4]. The PHDC generates disease specific reports, and for TB, collates data from laboratory sources (including smear, culture or Xpert MTB/RIF (Xpert)), pharmacy or clinical records, TB treatment registers or TB-specific elements recorded in electronic data systems at PHC or hospital level [4].

## Tygerberg and Khayelitsha sub districts

In 2019, the City of Cape Town had an estimated population of 4.1 million people, divided into 8 sub-districts, with sub-district population estimates of 661,555 (16.0%) for Tygerberg and 400,753 (9.7%) for Khayelitsha [5]. In 2019, the City of Cape Town reported 24,482 TB patients in the routine drug-susceptible TB treatment register (TIER.Net); an estimated case notification rate of 597/100,000. Tygerberg reported 3,447 (14.1%) and Khayelitsha 4,310 (17.6%) of these patients. Cape Town had an estimated antenatal HIV prevalence of 20.9% in 2017 [6]. There is a single district level hospital in Khayelitsha sub-district (Khayelitsha District Hospital) and three hospitals in Tygerberg sub-district including a tertiary (Tygerberg Hospital), a district (Karl Bremer Hospital) and a mental health hospital.

## Statistical analysis

Table 1 specifies the study definitions. To ascertain linkage to care and vital status, we assessed all routinely diagnosed TB patients 7 months after diagnosis. For all TB patients who did not link to TB services, the vital status was verified in the South African population register using personal identifiers as recorded in the PHDC. Descriptive statistics of demographic and clinical variables which were routinely collected by services were made available for all TB patients and stratified by ILTFU. ILTFU TB patients were stratified by the level of care where diagnosis was made (hospital or PHC) and by vital status. A logistic regression model was developed to assess predictors of mortality among all TB patients and a Cox regression model was used to evaluate risk factors for mortality among ILTFU TB patients. Following univariate analyses, predictors were added incrementally, observing the change in significance of the likelihood ratio test of each model, to produce final adjusted models. The time to death was calculated using the difference between the date of death and that of the TB diagnosis. Survival analysis was also completed for ILTFU TB patients, using Kaplan Meier curves. SAS software, Version 9.4. Copyright © 2002–2012 SAS Institute Inc., Cary, North Carolina, USA was used for data analysis.

## Ethics

The Health Research Ethics Committee of Stellenbosch University (N18/07/069) approved the study. Approvals were also received from the Western Cape Department of Health (WC_201808_034) and the City of Cape Town Health Directorate (8053). A waiver of informed consent was granted as this study analysed patients diagnosed and treated within routine health services.

**Table 1. Definitions used in the study.**

| Term | Definition |
| --- | --- |
| Initial loss to follow up (ILTFU) | TB patients who did not link with a TB treatment facility (PHC facility or TB hospital) in the province within 30 days of their TB diagnosis for TB treatment, regardless of whether they had initiated their TB treatment in hospital. Linkage to a TB treatment facility included attendance for TB treatment at a PHC facility or admission to a TB hospital following the TB diagnosis. TB patients defined as being ILTFU included those diagnosed at a TB treatment facility who neither linked to care at a different TB treatment facility nor returned to care at the facility where they were diagnosed and patients who had died before their linkage to TB care. |
| Death | Any patient reported to have died from any cause following the diagnosis of TB. The underlying cause of death was not evaluated, consistent with South African and global definitions of TB death [27, 28]. To ascertain death, we searched the provincial health data centre and the South African population register using personal identifiers for any recorded deaths up to 7 months following the diagnosis of TB. |
| No recorded attendance at PHC | Patients diagnosed with TB at a hospital who had no record of any attendance at any public sector PHC services prior to, and/or during the current diagnostic episode. |
| Bacteriologically confirmed | Individuals with positive smear, culture, LPA (line probe assay) or Xpert results reported by the National Health Laboratory Service. |
| Clinically diagnosed | Patients with a TB diagnosis generated from any source other than laboratory data |
| Pulmonary TB | Site of disease included any pulmonary involvement |
| Extrapulmonary TB | Site of disease exclusively affecting any organ other than the lung parenchyma |
| On antiretroviral therapy (ART) | Patients who had documented evidence of ART for the treatment of HIV regardless of the timing. This includes patients on ART prior to the diagnosis of TB and those started on ART after the diagnosis of TB |

ART: antiretroviral therapy; ILTFU: initial loss to follow up; PHC: primary health care; TB: tuberculosis.

## Results

Between 1 October 2018 and 31 March 2020, 14,471 individuals were diagnosed with TB. Fourteen patients diagnosed at a mental health hospital and 721 patients with drug-resistant TB were excluded. Of the 13,736 patients included, 74.3% (10,208/13,736) were diagnosed at a PHC facility. Overall, ILTFU was 20.0% (2,742/13,736) (Fig 1); 11.8% (1,201/10,208) in those diagnosed at PHC facilities and 43.7% (1,541/3,528) for those diagnosed in hospital (Fig 2). Bacteriological confirmation was reported in 66.2% (9,100/13,736) of TB patients.

### Early mortality

Mortality was 17.1% (468/2,742) in ILTFU patients and 3.3% (365/10,994) in TB patients who linked to TB services within 30 days of their diagnosis. Of ILTFU TB patients who died, 69.7% (326/468) died within 30 days of their TB diagnosis (Fig 1). The median time to death was 13 (IQR: 5–42) days for ILTFU TB patients and 59 (IQR: 29–109) days for patients who had linked within 30 days of diagnosis. Of all ILTFU TB patients, 82.7% (2,267/2,742) were ≥15 years of age; 76.7% (2,104/2,742) had bacteriologically confirmed TB; 43.4% (1,189/2,742) were HIV-positive with 80.8% (961/1,189) on antiretroviral therapy (ART); 58.6% (1,606/2,742) started TB treatment; and 56.2% (1,541/2,742) were diagnosed in hospital (Table 2). Of deceased ILTFU TB patients, 85.5% (400/468) occurred among those diagnosed in hospital; 94.2% (441/468) among adults; and 69.7% (326/468) within 30 days of the TB diagnosis (Table 3). Of ILTFU patients, 5.7% (68/1,201) of those diagnosed at PHC facilities died; 26.0% (400/1,541) of those diagnosed in hospital died (Table 3); and 21% (221/1,055) of those diagnosed in hospital who started TB treatment died (Table 4). Among those diagnosed in hospital who started TB treatment in hospital but never linked to ongoing care 27% (209/786) died. Of

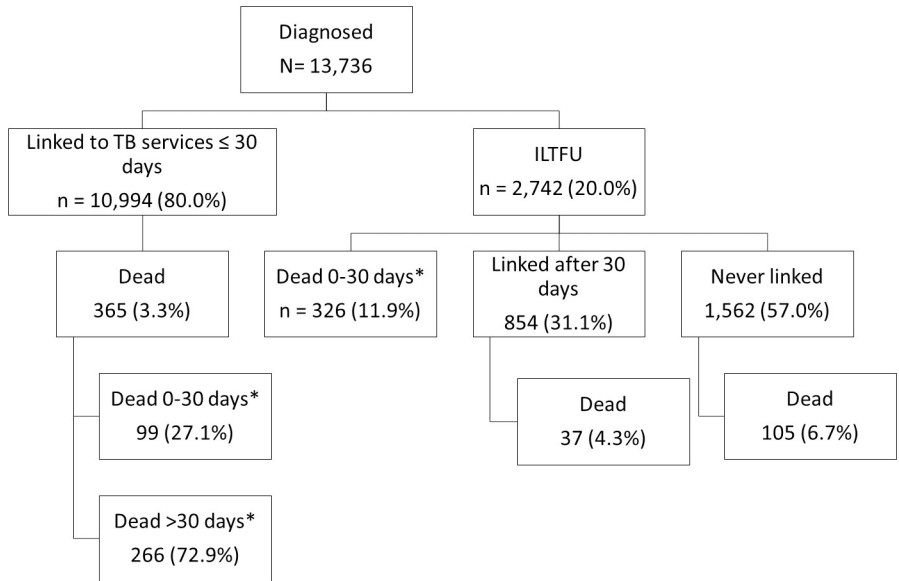

**Fig 1. Overview of linkage to TB care and death of patients diagnosed with TB in Cape Town, South Africa, October 2018-March 2020 (n = 13,736).** *proportions do not denote a case fatality ratio but the proportion of the deaths to have occurred based on the timing of death. ILTFU: initial loss to follow up.

ILTFU TB patients, 16.1% (441/2,742) had no recorded attendance at any PHC facility and 44.2% (195/441) of them died (S1 Table in S1 File provides a detailed profile of this group). Pregnancy was documented in 5.1% (46/904) of females (12–55 years old) who were ILTFU and 17.4% (8/46) of ILTFU pregnant women died.

## Predictors of mortality

Compared to TB patients diagnosed at PHC facilities who started TB treatment and were linked to TB services, TB patients diagnosed in hospital had an increased odds of mortality. TB

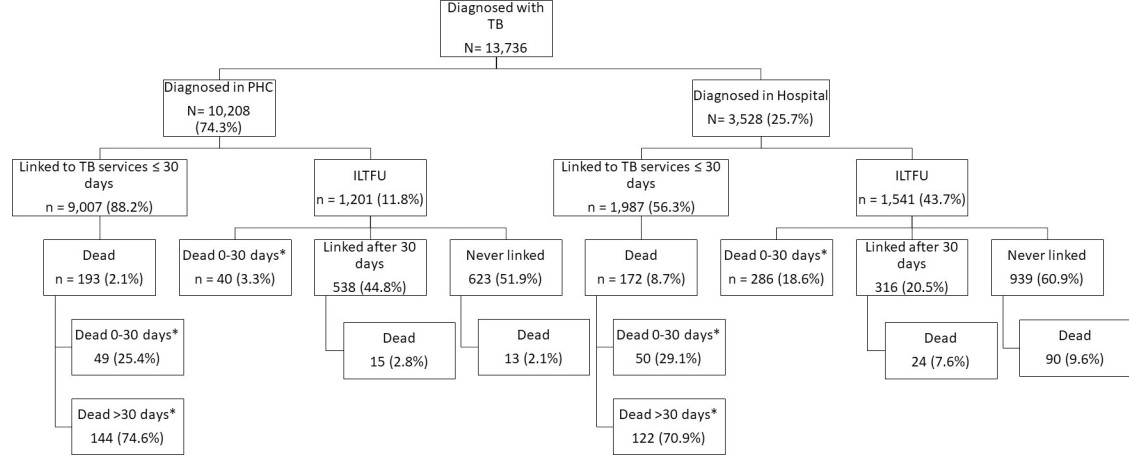

**Fig 2. Overview of linkage to TB care and death of patients, stratified by level of care at which TB diagnosis was made, Cape Town, South Africa, October 2018-March 2020 (n = 13,736).** *proportions do not denote a case fatality ratio but the proportion of the deaths to have occurred based on the timing of death. ILTFU: initial loss to follow up; PHC: primary health care.

**Table 2. Overall clinical and demographic characteristics of patients diagnosed with TB, stratified by linkage to TB services at 30 days, Cape Town, South Africa, October 2018-March 2020 (n = 13,736).**

| | | Linked to TB services ≤30 days | | Initial loss to follow up | |
|---|---|---|---|---|---|
| | | **10,994** | *col%* | **2,742** | *col%* |
| Age category | 0–4 years | 652 | 5.9 | 302 | 11.0 |
| missing = 1 | 5–14 years | 374 | 3.4 | 173 | 6.3 |
| | 15–24 years | 1,553 | 14.1 | 254 | 9.3 |
| | 25–34 years | 2,953 | 26.9 | 646 | 23.6 |
| | 35–44 years | 2,631 | 23.9 | 548 | 20.0 |
| | 45–54 years | 1,627 | 14.8 | 421 | 15.4 |
| | 55–64 years | 834 | 7.6 | 248 | 9.0 |
| | 65+ years | 369 | 3.4 | 150 | 5.5 |
| Sex | Female | 4,839 | 44.0 | 1,316 | 48.0 |
| missing = 25 | Male | 6,134 | 55.8 | 1,422 | 51.9 |
| HIV status | HIV+ no ART | 307 | 2.8 | 228 | 8.3 |
| | HIV+ on ART | 4,729 | 43.0 | 961 | 35.0 |
| | HIV- | 5,958 | 54.2 | 1,553 | 56.6 |
| Diagnostic method | Clinically diagnosed | 3,998 | 36.4 | 638 | 23.3 |
| | Bacteriologically confirmed | 6,996 | 63.6 | 2,104 | 76.7 |
| Site of disease | EPTB | 1,472 | 13.4 | 421 | 15.4 |
| | PTB | 5,500 | 50.0 | 278 | 10.1 |
| | Site not specified | 4,022 | 36.6 | 2,043 | 74.5 |
| Diabetes | No | 10,334 | 94.0 | 2,545 | 92.8 |
| | Yes | 660 | 6.0 | 197 | 7.2 |
| Level of care where diagnosis was made | Primary Health Care | 9,007 | 81.9 | 1,201 | 43.8 |
| | Hospital | 1,987 | 18.1 | 1,541 | 56.2 |
| TB treatment started | No | 1 | 0.0 | 1,136 | 41.4 |
| | Yes | 10,993 | 100.0 | 1,606 | 58.6 |
| Level of care of diagnosis and treatment status | Hospital diagnosis & started on treatment | 1,987 | 18.1 | 1,055 | 38.5 |
| | Hospital diagnosis not started on treatment | 0 | 0.0 | 486 | 17.7 |
| | PHC diagnosis & started on treatment | 9,006 | 81.9 | 551 | 20.1 |
| | PHC diagnosis not started on treatment | 1* | 0.0 | 650 | 23.7 |
| Dead | No | 10,629 | 96.7 | 2,274 | 82.9 |
| | Yes | 365 | 3.3 | 468 | 17.1 |
| Timing of death | ≤30 days | 99 | 27.1 | 326 | 69.7 |
| | >30 days | 266 | 72.9 | 142 | 30.3 |

ART: Antiretroviral therapy; DST: drug susceptibility test; EPTB: extra pulmonary TB; PTB: pulmonary TB; TB: tuberculosis; +: positive; -: negative.

*1 patient was diagnosed at a PHC facility and admitted to a TB hospital and therefore met the definition of linked to TB services. This patient died with no electronic evidence of TB treatment and is therefore classified as *'PHC diagnosis not started on treatment'*.

patients diagnosed in hospital who started their TB treatment and were linked to ongoing TB services had an aOR: 3.29 (95%CI: 2.62–4.13), while those who started their TB treatment and were ILTFU had an aOR: 9.53 (95%CI: 7.56–12.02), and those who did not start their TB treatment and were ILTFU had an aOR: 28.66 (95%CI: 21.53–38.15) (Table 4). ILTFU following a TB diagnosis in PHC facilities also increased the odds of mortality, and this was higher in those who did not start their TB treatment, aOR: 3.48 (95%CI:2.45–4.94) as opposed to those who had been started on TB treatment, aOR: 1.47 (95%CI: 0.9–2.41) (Table 4).

**Table 3. Mortality amongst initial loss to follow up TB patients, stratified by level of care at which diagnosis was made, Cape Town, South Africa, October 2018-March 2020 (n = 2,742).**

| | | Hospital | | | Primary health care | | |
|---|---|---|---|---|---|---|---|
| | | ILTFU | deaths | row% | ILTFU | deaths | row% |
| | | **1,541** | **400** | **26.0** | **1,201** | **68** | **5.7** |
| Age category | 0–4 years | 293 | 22 | 7.5 | 9 | 0 | 0.0 |
| | 5–14 years | 89 | 5 | 5.6 | 84 | 0 | 0.0 |
| | 15–24 years | 105 | 27 | 25.7 | 149 | 2 | 1.3 |
| | 25–34 years | 325 | 80 | 24.6 | 321 | 14 | 4.4 |
| | 35–44 years | 263 | 81 | 30.8 | 285 | 24 | 8.4 |
| | 45–54 years | 205 | 65 | 31.7 | 216 | 14 | 6.5 |
| | 55–64 years | 146 | 64 | 43.8 | 102 | 9 | 8.8 |
| | 65+ years | 115 | 56 | 48.7 | 35 | 5 | 14.3 |
| Sex | Female | 756 | 204 | 27.0 | 560 | 33 | 5.9 |
| missing = 4 | Male | 784 | 196 | 25.0 | 638 | 35 | 5.5 |
| HIV status | HIV+ no ART | 175 | 75 | 42.9 | 53 | 7 | 13.2 |
| | HIV+ on ART | 449 | 145 | 32.3 | 512 | 42 | 8.2 |
| | HIV- | 917 | 180 | 19.6 | 636 | 19 | 3.0 |
| Diagnostic method | Clinically diagnosed | 555 | 137 | 24.7 | 83 | 8 | 9.6 |
| | Bacteriologically confirmed | 986 | 263 | 26.7 | 1,118 | 60 | 5.4 |
| Site of disease | EPTB | 362 | 95 | 26.2 | 59 | 10 | 16.9 |
| | PTB | 81 | 1 | 1.2 | 197 | 2 | 1.0 |
| | Site not specified | 1,098 | 304 | 27.7 | 945 | 56 | 5.9 |
| Diabetes | No | 1,402 | 347 | 24.8 | 1,143 | 65 | 5.7 |
| | Yes | 139 | 53 | 38.1 | 58 | 3 | 5.2 |
| TB treatment started | No | 486 | 179 | 36.8 | 650 | 49 | 7.5 |
| | Yes | 1,055 | 221 | 20.9 | 551 | 19 | 3.4 |
| PHC record | No recorded attendance at PHC | 441 | 195 | 44.2 | | | |
| | Attended PHC before and/or during TB episode | 1,100 | 205 | 18.6 | 1,201 | 68 | 5.7 |
| Timing of death** | ≤30 days | 400 | 286 | 71.5 | 68 | 40 | 58.8 |
| | >30 days | 400 | 114 | 28.5 | 68 | 28 | 41.2 |

** for timing of death, n = number of deaths, not total number of TB patients.

ART: Antiretroviral therapy; DST: drug susceptibility test; EPTB: extra pulmonary TB; PTB: pulmonary TB; TB: tuberculosis; +: positive; -: negative.

## Survival analysis

Using Kaplan Meier survival curves for ILTFU TB patients, the cumulative mortality was 34% at 15 days, and 46% at 30 days after a TB diagnosis if HIV-positive and not on ART;14% and 18% if HIV-positive on ART; and 13% and 18% if HIV-negative, with the overall curves for HIV-negative and HIV-positive patients on ART being similar (Fig 3). Among hospital-diagnosed ILTFU TB patients who did not start TB treatment, there was an early and steep decline in survival; the cumulative mortality was 61% at 15 days and 71% at 30 days after TB diagnosis (Fig 4). ILTFU TB patients who had no record of attendance at a PHC facility had a cumulative mortality of 43% at 15 days and 57% at 30 days, compared to those who had a record of a PHC facility attendance before the TB diagnosis and/or during the TB episode, who had a cumulative mortality of 10% at 15 days, and 13% at 30 days (Fig 5). In the multivariable Cox regression model, ILTFU TB patients diagnosed at hospital had an increased hazard of mortality, which was five times higher in the absence of TB treatment initiation, aHR: 34.56 (95%CI:

**Table 4. Univariate and multivariable logistic regression model with predictors of death among diagnosed TB patients, Cape Town, South Africa, October 2018-March 2020 (n = 13,736\*).**

| | | Total | Deaths | CFR% | OR (95%CI) | aOR (95%CI) |
|---|---|---|---|---|---|---|
| Age category | 0–4 years | 954 | 23 | 2.4 | 2.23 (0.90–5.5) | 1.45 (0.58–3.63) |
| | 5–14 years | 547 | 6 | 1.1 | reference | |
| | 15–24 years | 1,807 | 38 | 2.1 | 1.94 (0.81–4.6) | 3.49 (1.44–8.43) |
| | 25–34 years | 3,599 | 182 | 5.1 | 4.80 (2.18–10.87) | 5.24 (2.27–12.09) |
| | 35–44 years | 3,179 | 215 | 6.8 | 6.54 (2.89–14.78) | 7.23 (3.13–16.69) |
| | 45–54 years | 2,048 | 137 | 6.7 | 6.46 (2.84–14.71) | 6.81 (2.93–15.82) |
| | 55–64 years | 1,028 | 129 | 11.9 | 12.20 (5.35–27.83) | 16.12 (6.93–37.52) |
| | 65+ years | 519 | 103 | 19.9 | 22.31 (9.70–51.30) | 27.43 (11.65–64.6) |
| Sex | Male | 7,556 | 429 | 5.7 | reference | |
| | Female | 6,155 | 504 | 6.6 | 1.17 (1.01–1.34) | 0.98 (0.84–1.15) |
| HIV | HIV- | 7,511 | 312 | 4.2 | reference | |
| | HIV+ no ART | 5,690 | 425 | 7.5 | 5.05 (3.94–6.47) | 3.11 (2.31–4.18) |
| | HIV+ on ART | 535 | 96 | 17.9 | 1.86 (1.60–2.17) | 2.24 (1.85–2.73) |
| Diagnostic method\*\* | Bacteriologically confirmed | 9,100 | 547 | 6.0 | reference | |
| | Clinically diagnosed | 4,636 | 286 | 6.2 | 1.03 (0.89–1.19) | 1.19 (1–1.42) |
| Diabetes | No | 12,879 | 739 | 5.7 | reference | |
| | Yes | 857 | 94 | 11.0 | 2.02 (1.61–2.54) | 1.15 (0.87–1.51) |
| Site of disease | PTB | 5,778 | 109 | 1.9 | reference | |
| | EPTB | 1,893 | 245 | 12.9 | 7.73 (6.13–9.75) | 3.00 (2.31–3.9) |
| | Site not specified | 6,065 | 479 | 7.9 | 4.46 (3.61–5.51) | 1.75 (1.37–2.23) |
| Level of care of diagnosis\*\* | Primary Health Care (PHC) | 10,208 | 261 | 2.6 | Reference | |
| | Hospital | 3,528 | 572 | 16.2 | 7.38 (6.33–8.59) | |
| TB treatment started\*\* | No | 1,137 | 229 | 20.1 | reference | |
| | Yes | 12,599 | 604 | 4.8 | 0.2 (0.17–0.24) | |
| Initial loss to follow up (ILTFU)\*\* | No | 10,994 | 365 | 3.3 | reference | |
| | Yes | 2,742 | 468 | 17.1 | 5.99 (5.19–6.92) | 2.47 (2.01–3.03) |
| Level of care of diagnosis, treatment status, and ILTFU | Diagnosed PHC, Treatment started, Linked | 9,006 | 192 | 2.1 | reference | |
| | Diagnosed PHC, Treatment started, ILTFU | 551 | 19 | 3.5 | 1.64 (1.02–2.65) | 1.47 (0.9–2.41) |
| | Diagnosed PHC, Treatment not started, ILTFU | 651 | 50 | 7.7 | 3.82 (2.77–5.27) | 3.48 (2.45–4.94) |
| | Diagnosed hospital, Treatment started, Linked | 1,987 | 172 | 8.7 | 4.35 (3.52–5.38) | 3.29 (2.62–4.13) |
| | Diagnosed hospital, Treatment started, ILTFU | 1,055 | 221 | 21.0 | 12.16 (9.9–14.95) | 9.53 (7.56–12.02) |
| | Diagnosed hospital, Treatment not started, ILTFU | 486 | 179 | 36.8 | 26.77 (21.2–33.8) | 28.66 (21.53–38.15) |

ART: Antiretroviral therapy; aOR: adjusted odds ratio; EPTB: extra pulmonary TB; ILTFU: initial loss to follow up; OR: odds ratio; PHC: primary health care; PTB: pulmonary TB; TB: tuberculosis; +: positive; -: negative.

\*there were 13,736 TB patients diagnosed but 1 had a missing age and 25 did not specify sex as male or female. In the univariate analyses for age n = 13,735 and for sex n = 13,711. In the final multivariable model n = 13,710.

\*\* Variables not included in multivariable model due to collinearity. Level of care of diagnosis, TB treatment started, and Initial loss to follow up collinear with Level of care of diagnosis, treatment status, and ILTFU.

21.26–56.17) for those not started on treatment compared to those started on TB treatment (aHR: 6.98 (95%CI: 4.3–11.32)) (S2 Table in S1 File). Not starting TB treatment following a diagnosis at a PHC facility also increased the hazard of mortality in the final adjusted model, aHR: 14.87 (95%CI: 8.64–25.58). (S2 Table in S1 File).

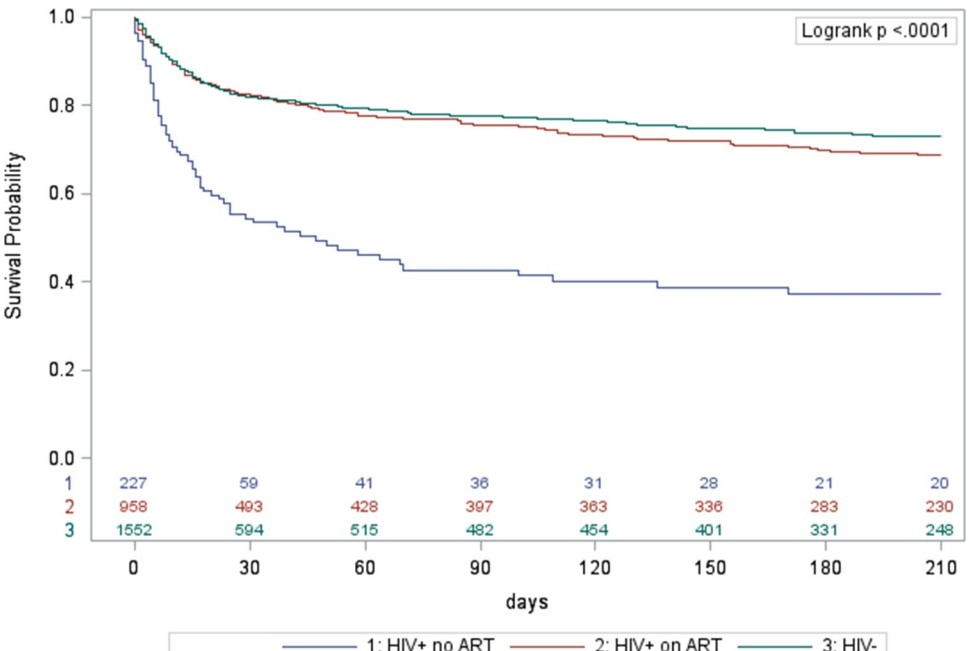

**Fig 3. Kaplan Meier survival curves for initial loss to follow up TB patients, Cape Town, South Africa, October 2018-March 2020, stratified by HIV status.** ART: antiretroviral therapy; HIV-: HIV-negative; HIV+: HIV-positive; TB: tuberculosis.

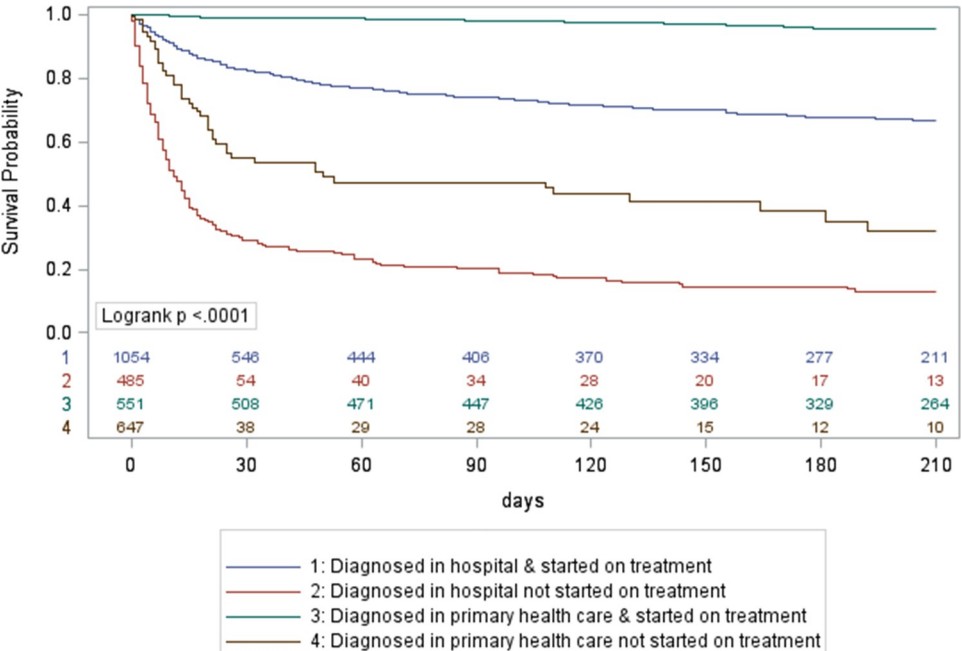

**Fig 4. Kaplan Meier survival curves for initial loss to follow up TB patients, Cape Town, South Africa, October 2018-March 2020, stratified by level of care of diagnosis and treatment initiation status.** TB: tuberculosis.

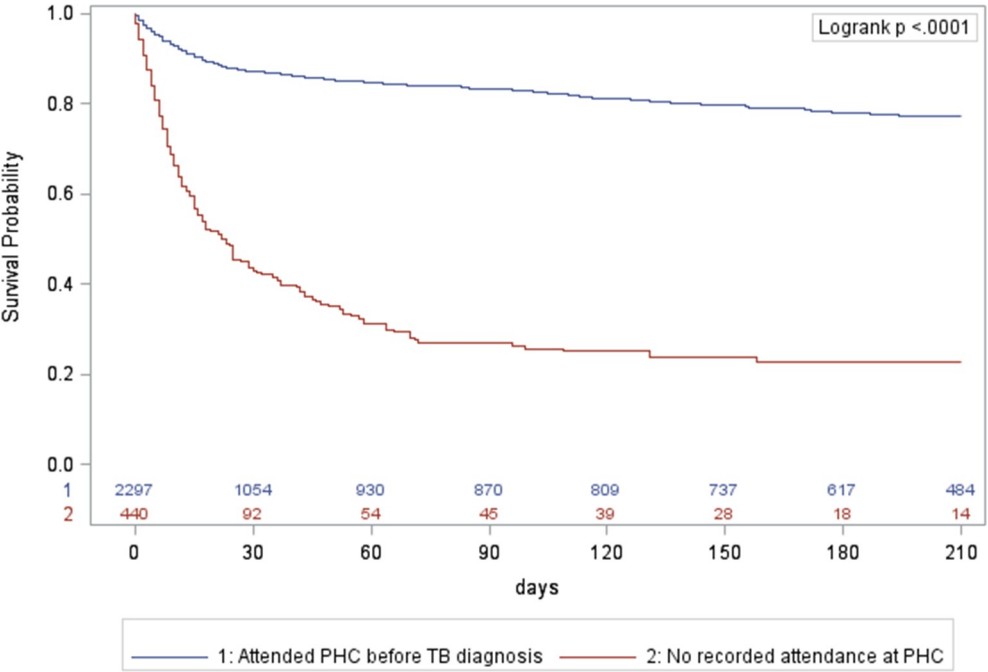

**Fig 5. Kaplan Meier survival curves for initial loss to follow up TB patients, Cape Town, South Africa, October 2018-March 2020, stratified by patients known to primary health care or hospital status.** PHC: primary health care; TB: tuberculosis.

## Discussion

Mortality before TB registration provides estimates of TB mortality in individuals who were diagnosed with TB but are not currently included in routine TB surveillance data. We were able to document the profile and impact of being ILTFU on mortality in a high burden TB setting where TB services are decentralized, focusing on diagnosis and treatment at the PHC level. Of diagnosed TB patients, 20.0% were ILTFU and of these, 17.1% died. This is in stark contrast to those TB patients who linked to care where 3.3% died. The proportion of ILTFU TB patients who died was considerably higher amongst those diagnosed in hospital compared to PHC facilities (26.0% vs. 5.7%). In multivariable analysis, older age, HIV positive status (with an even greater increase in the absence of ART), extrapulmonary TB, diagnosis in hospital (substantially higher in the absence of treatment initiation and ILTFU) increased the odds of dying within 7 months of a diagnosis of TB.

   In recent work, the proportions of pre-treatment loss to follow up amongst TB patients were similar (~20%) in Uganda [7], South Africa [8], Zimbabwe [9], and India [10] but the estimates of mortality differed. In South Africa, 2% died before starting treatment but mortality was not ascertained for the group who were lost to follow up before treatment [8]. In Uganda, 22.4% of pre-treatment loss to follow up TB patients died, but mortality was only ascertained in 49 out of 100 patients with a positive Xpert results who were not initiated on TB treatment within two weeks of diagnosis [7]. In Zimbabwe 49.6% (252/508) of the pre-treatment loss to follow up TB patients died between 2012–2016 [9]. In India, 27.6% (21/76) died before treatment or registration [10]. Definitions used in these studies differed, with most studies measuring mortality before treatment. The Indian study used a similar definition to our study and estimated mortality before treatment or registration. Developing consensus definitions for TB mortality before treatment initiation or registration would be a useful step

towards interventions to reduce mortality and improve estimates of TB mortality. Considering TB patients that are diagnosed in hospitals require linkage to TB treatment facilities, definitions which include losses before the definitive linkage with a TB treatment facility need to be considered. Where National TB programs continue to focus on TB mortality as part of outcome indicators of TB treatment cohorts, the true burden of TB mortality will continue to be under-reported and critical opportunities to identify and address underlying determinants of TB mortality will remain unaddressed.

Our estimate of 13 (IQR: 5–42) days to death from diagnosis among ILTFU TB patients is shorter than previous work which reported median time from diagnosis to death of 3.5 weeks [3] and 22 days [7]. This may be related to our focus on ILTFU regardless of treatment status or ascertaining more mortality in a much larger population of ILTFU patients. Early mortality likely reflects more severe TB disease at the time of diagnosis and the increased odds of mortality we demonstrated in patients diagnosed in hospital compared to PHC facilities, further supports this. High numbers of patients presenting to hospitals with advanced TB disease might reflect barriers in accessing primary health care at an earlier stage of TB disease. This may highlight opportunities for earlier TB case detection, before hospital admission, and could include active case finding in communities using targeted approaches, such as radiological evaluation, or screening and testing of high-risk populations e.g. TB contacts, previously treated TB patients, and people living with HIV. Of the ILTFU TB patients who died within 30 days of diagnosis, 87.7% (286/326) were diagnosed in hospital, and cumulative mortality was the greatest in hospital-diagnosed patients who did not start TB treatment. The use of urinary lipoarabinomannan (LAM) for the diagnosis of TB in HIV-positive patients, presents an opportunity for the rapid detection of TB and stratification of patients at high risk of mortality, as LAM in urine is associated with increased risk of mortality during TB treatment [11]. While it is unclear if urinary LAM serves as a marker of disease severity or a compromise in host immunity [11], it would be opportune to consider LAM results in the package of care, including further admission to TB hospitals or further support via PHC services once discharged, for these patients at high risk of mortality. Patient factors, health seeking behaviour and acceptability and accessibility of health services are all important in TB diagnostic delays in systematic reviews [12–14]. Among ILTFU TB patients, 44.2% of those whose first presentation to health services was at hospital died. Interventions to change health seeking behaviour, encouraging patients to seek care earlier at PHC are needed. Additional mechanisms to flag such vulnerable patients upon hospital admission could present an opportunity for adjuvant, individualised TB treatment during the hospital admission. Treatment initiation reduced mortality in the univariate model and in the multivariable model we considered the level of care of diagnosis, treatment status, and linkage to TB care. TB patients diagnosed in hospital had high odds of mortality, with odds three times higher in those who were ILTFU and a further three times higher in those who had not started treatment and were ILTFU. The importance of rapid treatment initiation is unequivocal for the patient [15–17] and reducing transmission [18, 19]. Our findings highlight that treatment initiation remains important and needs to occur with improved linkage to ensure ongoing TB care, especially for patients moving between different levels of healthcare. In our setting, the registration of TB patients in hospital at the time of diagnosis can take place through the integrated PHDC. However, in other settings, the registration of TB patients in-hospital at the time of diagnosis is an important intervention that is required. The inclusion of these patients in routine programmatic reporting is essential to ensure that early on-treatment mortality is recorded and to guide TB case-finding interventions to ensure earlier care and prevent mortality. We have shown that following this registration, TB treatment initiation and linkage to TB services for continued treatment and care remain critical steps to further reduce TB mortality.

Among ILTFU TB patients, older age categories had an increased odds of death. This is consistent with risk factors for mortality during TB treatment [20] and may be reflective of underlying clinical factors, co-morbidity and additional reasons for death in older patients. The effect of HIV and ART on TB mortality during TB treatment has been well documented [20–23]. Here we have quantified the effect of HIV on TB mortality prior to TB registration. We have demonstrated the reduction in odds of mortality with ART among HIV-positive TB patients and the improvement in survival among ILTFU TB patients who were HIV-positive and on ART. The odds of mortality increased with a diagnosis of diabetes in univariate analyses, but this effect was not maintained in the multivariable model. This may relate to the higher proportion of TB patients with diabetes diagnosed at hospital level (8.7%) compared to the proportion of TB patients with diabetes diagnosed at PHC facilities (5.3%). Further work examining the effect of diabetes and other co-morbidities on ILTFU and mortality is needed.

We documented a large proportion of children 17.3% (475/2,742) among ILTFU TB patients, disproportionately high compared to the total disease burden of children in this study [10.9% (1,501/13,736)]. In children, linkage to care requires adult support and further engagement with parents on the factors limiting linkage are required. Children accounted for a smaller proportion of ILTFU TB patients who died 5.8% (27/468) with all recorded deaths amongst children diagnosed in hospital. In earlier work from South Africa, children with TB meningitis or death prior to referral were less likely to be documented in the TB register [24, 25]. In our work we were able to demonstrate the increased odds of mortality in patients with extrapulmonary TB but did not look at the specific sites of disease or distribution by age. Importantly, the 27 children who were ILTFU and died following a diagnosis of TB in hospital would not be reported in any routine TB register and their inclusion in routine reporting is essential to improve estimates of paediatric TB mortality in this setting.

A strength of this study is that we were able to identify the majority of diagnosed TB patients and were not restricted to laboratory confirmed TB patients. We were also not restricted to matching within TB treatment registers and were able to track attendance at PHC facilities or admissions at TB hospitals across the province. Previous studies based on paper-based treatment registers [7–10] were not able to track patients who returned to care after an initial test and may have over-estimated ILTFU. By using the electronic sources with a unique identifier, we were able to identify patients who returned for repeat testing and accessed treatment at any public facility in the Western Cape Province. This is a strength of the study as we were able to more accurately ascertain those patients who were classified as ILTFU and never returned to care. Searching for ILTFU TB patients in the population vital statistics register is a further strength, as earlier work showed that the South African vital registration system has a high level of completeness for mortality data [26]. We acknowledge that the search was based on individual patient details and that the difficulties of matching using names and surnames with missing identification numbers may have under ascertained mortality. We excluded patients with proven drug-resistant TB in this analysis and further work evaluating ILTFU and mortality among patients with drug-resistant TB is needed. We acknowledge that we did not distinguish the exact time to treatment initiation, or the place of TB treatment initiation and that the group of TB patients who were diagnosed in hospital and had started treatment and were ILTFU included both a group of patients who never linked to care as well as patients who did link to TB care after 30 days. Additional work to evaluate the level of care at which TB treatment is initiated, time to treatment initiation, the duration of the initial treatment provided, and the time taken to link to TB treatment services after initial hospital admission is needed. The most significant limitation is the inability to assess reverse causality between ILTFU and early mortality. While we have documented the increased risk of death among ILTFU TB patients, this might include patients who were lost to follow-up because of their

early and unreported death. Further prospective work is needed to evaluate the impact of interventions to reduce ILTFU, and the impact on TB mortality.

The goal of zero TB deaths requires the careful evaluation of death across the TB care cascade. Failing to go beyond the evaluation of case fatality ratios in individuals who are on TB treatment and have been registered will continue to substantially under report the burden of TB deaths. We found that one in five patients diagnosed with TB did not link to the routine TB services within 30 days of their diagnosis; 12% of whom died within 30 days and a further 5% died after 30 days, reflecting a staggering 17% mortality. Individuals diagnosed with TB in hospitals are a modest proportion of all diagnosed TB but contribute excessively to TB mortality, especially early mortality among those ILTFU. While a hospital-based diagnosis of TB likely reflects more severe TB disease, patients diagnosed at hospital, especially those who present to a hospital for their first engagement with health services, present an opportunity to identify TB patients at highest risk of early and overall mortality. The provision of additional treatment interventions, support, and facilitators of linkage to care for adults and children diagnosed with TB in hospital are needed. The registration of TB patients in-hospital at the time of their diagnosis is an important intervention to support linkage to TB services to reduce early mortality and ensure that when occurring, early on-treatment mortality is recorded.

## Supporting information

**S1 Checklist. The RECORD statement–checklist of items, extended from the STROBE statement, that should be reported in observational studies using routinely collected health data.**
(DOCX)

**S1 File.**
(PDF)

## Acknowledgments

We wish to acknowledge our implementing partners; the University of Cape Town and the Centre for Infectious Disease Epidemiology (CIDER) in the Western Cape Province. We further acknowledge the staff at the Western Cape Provincial Health Data Centre (PHDC) for their invaluable assistance. We highly appreciate input from the health staff at the provincial, district and sub-district health offices and those at the facilities in which the study was implemented. Our sincere appreciation to all TB patients.

## Author Contributions

**Conceptualization:** Sue-Ann Meehan, Karen Du Preez, Florian M. Marx, Alex Welte, Pren Naidoo, Anneke C. Hesseling.

**Data curation:** Muhammad Osman, Sue-Ann Meehan, Arne von Delft, Rory Dunbar, Andrew Boulle.

**Formal analysis:** Muhammad Osman, Sue-Ann Meehan, Arne von Delft, Florian M. Marx, Andrew Boulle, Alex Welte, Pren Naidoo, Anneke C. Hesseling.

**Funding acquisition:** Muhammad Osman, Sue-Ann Meehan, Anneke C. Hesseling.

**Investigation:** Muhammad Osman, Sue-Ann Meehan.

**Methodology:** Muhammad Osman, Sue-Ann Meehan, Arne von Delft, Karen Du Preez, Rory Dunbar, Florian M. Marx, Andrew Boulle, Alex Welte, Pren Naidoo, Anneke C. Hesseling.

**Project administration:** Muhammad Osman, Sue-Ann Meehan.

**Resources:** Anneke C. Hesseling.

**Supervision:** Muhammad Osman, Andrew Boulle, Alex Welte, Anneke C. Hesseling.

**Validation:** Arne von Delft.

**Writing – original draft:** Muhammad Osman, Arne von Delft.

**Writing – review & editing:** Muhammad Osman, Sue-Ann Meehan, Arne von Delft, Karen Du Preez, Rory Dunbar, Florian M. Marx, Andrew Boulle, Alex Welte, Pren Naidoo, Anneke C. Hesseling.

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
