## [Decision Letter · Decision Letter 0]

22 Mar 2021

PONE-D-20-40984

Early mortality in tuberculosis patients initially lost to follow up following diagnosis at hospital or primary health care facility

PLOS ONE

Dear Dr. Osman,

Thank you for submitting your manuscript to PLOS ONE. After careful consideration, we feel that it has merit but does not fully meet PLOS ONE’s publication criteria as it currently stands. Therefore, we invite you to submit a revised version of the manuscript that addresses the points raised during the review process.

I would ask you to address the comments from the three reviewers. I do agree with the general comment from reviewer #3 that the conflation of the two slightly different groups does mean the manuscript can be difficult to follow at times. I do though think it’s important that all the information is included, because this poor linkage to follow-up care after hospital treatment initiation is an under-recognised problem and you are reporting very valuable information.

Given that this manuscript is based on observational routinely-collected health data, please could you ensure that reporting is in line with the RECORD statement (http://www.record-statement.org/) and submit a completed checklist with the revised manuscript.

We look forward to receiving your revised manuscript.

Kind regards,

Richard John Lessells, BSc, MBChB, MRCP, DTM&H, DipHIVMed, PhD

Academic Editor

PLOS ONE

Journal Requirements:

2. Please ensure you have included the registration number for the interventional study/ clinical trial referenced in the manuscript.

Reviewers' comments:

Reviewer's Responses to Questions

**Comments to the Author**

1. Is the manuscript technically sound, and do the data support the conclusions?

Reviewer #1: Yes

Reviewer #2: Yes

Reviewer #3: Yes

2. Has the statistical analysis been performed appropriately and rigorously? 

Reviewer #1: Yes

Reviewer #2: Yes

Reviewer #3: Yes

3. Have the authors made all data underlying the findings in their manuscript fully available?

Reviewer #1: No

Reviewer #2: No

Reviewer #3: Yes

4. Is the manuscript presented in an intelligible fashion and written in standard English?

Reviewer #1: Yes

Reviewer #2: Yes

Reviewer #3: Yes

5. Review Comments to the Author

Reviewer #1: This manuscript reports early mortality in tuberculosis patients initially lost to follow up following diagnosis at hospital or primary health care facility in South Africa. In overall, the statistical analysis is adequate and meet the goal of the planed data analysis. In addition to Figure 3, similar to Tale 4, are there data available to investigate predictors associated with overall survival for initial loss to follow up TB patients with Cox regression models?

Reviewer #2: Thank you for asking me to review this manuscript. Pre-treatment loss to follow-up is an extremely important contributor to poor TB patient and programme outcomes, but little is known about the fate of individuals who are lost to care before starting treatment. This well-conducted and clearly reported study is therefore a very important addition to the literature. I have only a small number of issues that the authors should address.

1) I think in the introduction, important to note that people with TB who are lost to follow-up before treatment may “recycle” into care later, for example by being diagnosed again or treated at another clinic. This has been very difficult to estimate using (predominately paper-based) TB lab and treatment register studies. This study therefore provides an important addition to the literature by capturing such events, reducing over-estimation of the pre-treatment loss to follow-up rate.

2) As the primary outcome is mortality, could the authors provide some assurance of the completeness and accuracy of data linkage with the vital registration system? E.g. by referencing previous validation studies?

Minor points

1) Line 63: incidence probably better expressed per 100,000 population, or instead reword to be “an estimated 360,000 people developed active TB”.

2) Figure 3: could the authors put numbers at risk at each time point under the KM plots?

Reviewer #3: Reviewer Reports:

This study examined early mortality in tuberculosis patients initially lost to follow up following diagnosis in two provincial hospitals in Western Cape- South Africa. The study represents interesting findings and adds to the growing body of evidence on the true burden of TB associated mortality.

General Comment:

The paper includes, in its definition of ITLFU both true initial loss to follow-up (no documented TB treatment initiation within a defined time period) and early on-treatment loss to follow-up (documented TB treatment initiation but no documented continuation on TB treatment). The combination of these two definitions makes the paper difficult to read and the results difficult to follow.

The general recommendation would be for the paper to focus on true initial loss to follow-up (no documented TB treatment initiation within a defined time period) vs all those initiated on TB treatment. This eliminate the one of the significant limitations of this study which is the inability to assess reverse causality between ILTFU and early mortality among those who did initiate TB treatment within the recommended time.

Alternatively, the title could be changed to “Early mortality in patients diagnosed with tuberculosis in two provincial hospitals in Western Cape- South Africa”.

Methods:

The study includes both patients with drug susceptible TB (DSTB) and those with multidrug resistant TB (MDR TB) are mutually exclusive cohorts. DSB and MDRTB tend to have different paths to treatment initiation/registration. As seen in your data (Table 3), initial loss to follow-up was higher among MDR TB patients diagnosed at PHC than those diagnosed in hospital. In addition, MDR TB treatment is longer so a follow-up period of 7 months will not capture all mortality during TB treatment ( the definition used in this study). It is therefore not appropriate to combine both groups of patients in the same analysis. Recommendation would be to include only one group of patients for this analysis (DS TB)

As ILFU more commonly means patient who are not initiated on TB treatment within a set period of time, it seems more appropriate to include treatment initiation or the lack thereof in your adjusted model. If there is co-linearity between treatment initiation and healthcare level (and there is), it might be the better choice to include treatment initiation and leave out healthcare level. If this is not possible, please explain why.

Result:

There is no mention on the effect of other co-morbidities (apart from HIV) e.g. diabetes on mortality following TB diagnosis in the univariate or bivariate analysis.

Tables are crowded. Legibility could be improved by deleting the column marked total (farthest to the left) in Tables 2 and 3

Figure 3 is ineligible.

Discussion:

In a number of countries, patients are registered in the TB treatment register on initiation of TB treatment. This might be a recommendation this study makes so that early on-treatment mortality is recorded in the national registry and used to guide TB case-finding interventions

A number of studies -including prevalence surveys- show that a) patients seek care for TB symptoms multiple times before being diagnosed with TB and that b) a high proportion of patients who present to primary care facilities with signs and symptoms of TB do not get tested with. It is therefore likely that the presentation to a hospital with advanced TB disease partly represents a failure of the health system. Kindly include this in your discussion.

Line 204: the more appropriate representation of the results would be mortality was only ascertained in 49 out of 100 patients who were not initiated on TB treatment within two weeks of diagnosis or 49% of patients not initiated on TB treatment within two weeks of diagnosis.

Please provide references this statement in Lines 242 and 243 The importance of rapid treatment initiation is unequivocal for the patient and reducing transmission. There are many studies particularly among HIV+ patients that compared early to later TB treatment initiation.

6. PLOS authors have the option to publish the peer review history of their article (what does this mean?). If published, this will include your full peer review and any attached files.

Reviewer #1: No

Reviewer #2: **Yes: **Peter MacPherson

Reviewer #3: No

---

## [Author Response · Author response to Decision Letter 0]

23 Apr 2021

Dear Editor

Thank you for the opportunity to revise our manuscript and respond to the reviewers’ comments.

We have responded to each comment below. In the attached response to reviewers we have marked changes in red and have added the text from our revised manuscript in italics for ease of review. We appreciate the general supportive comments from the reviewers on the relevance of the work, the overall structure and the analysis.

Response to reviewers

I would ask you to address the comments from the three reviewers. I do agree with the general comment from reviewer #3 that the conflation of the two slightly different groups does mean the manuscript can be difficult to follow at times. I do though think it’s important that all the information is included, because this poor linkage to follow-up care after hospital treatment initiation is an under-recognised problem and you are reporting very valuable information.

Thank you to the editor for this comment. We have responded in detail to reviewer 3 and have provided the rationale for the definition used. We agree that linkage to care following TB treatment initiation in hospital is important in this setting and have opted to maintain this focus.

Given that this manuscript is based on observational routinely-collected health data, please could you ensure that reporting is in line with the RECORD statement (http://www.record-statement.org/) and submit a completed checklist with the revised manuscript.

We have now reported according to the RECORD statement. The completed RECORD statement is attached. 

We have revised the format and changes are marked.

2. Please ensure you have included the registration number for the interventional study/ clinical trial referenced in the manuscript.

The underlying study is not a trial but an observational operational research study using a quasi-experimental (before-after) design. In the underlying study we used a method of registration at hospitals and an alert and response system for routine programmatic personnel. The study does not have a trial registration number as it is not assigning groups or individuals to an intervention but is implementing health system strengthening activities as part of routine health services, without controls. We have updated the text to specify that this analysis was nested in operational research:

Line 110: This analysis was nested in an operational research study aimed at reducing ILTFU using health system strengthening initiatives in these two large sub-districts….

We have engaged with the Western Cape Government: Health Impact Assessment Unit, the custodians of the Provincial Health Data Centre and the data used in this study in South Africa. They have confirmed that data should not be made publicly available and have provided contact information for the Provincial Health Data Centre and the Director, who have provided assurance that data may be made available through the existing routine application process of data for research, provided all criteria for data access are met.

Data is not being made available

Included on page 38

5. Review Comments to the Author

Reviewer #1: This manuscript reports early mortality in tuberculosis patients initially lost to follow up following diagnosis at hospital or primary health care facility in South Africa. In overall, the statistical analysis is adequate and meet the goal of the planed data analysis. 

Thank you for this feedback.

In addition to Figure 3, similar to Table 4, are there data available to investigate predictors associated with overall survival for initial loss to follow up TB patients with Cox regression models?

This data is available. We have now completed univariate and multivariable Cox regression analyses restricted to TB patients who were ILTFU and have produced hazard ratios. We have included the results of the Cox regression as a supplementary table (S2 Table) which is referenced in the main text.

Line 268-274: In the multivariable Cox regression model, ILTFU TB patients diagnosed at hospital had an increased hazard of mortality, which was five times higher in the absence of TB treatment initiation, aHR: 34.56 (95%CI: 21.26-56.17) for those not started on treatment compared to those started on TB treatment (aHR: 6.98 (95%CI: 4.3-11.32)) (S2 Table). Not starting TB treatment following a diagnosis at a PHC facility also increased the hazard of mortality in the final adjusted model, aHR: 14.87 (95%CI: 8.64-25.58). (S2 Table)

Reviewer #2: Thank you for asking me to review this manuscript. Pre-treatment loss to follow-up is an extremely important contributor to poor TB patient and programme outcomes, but little is known about the fate of individuals who are lost to care before starting treatment. This well-conducted and clearly reported study is therefore a very important addition to the literature. I have only a small number of issues that the authors should address.

1) I think in the introduction, important to note that people with TB who are lost to follow-up before treatment may “recycle” into care later, for example by being diagnosed again or treated at another clinic. This has been very difficult to estimate using (predominately paper-based) TB lab and treatment register studies. This study therefore provides an important addition to the literature by capturing such events, reducing over-estimation of the pre-treatment loss to follow-up rate.

Thank you for this comment. We have updated the introduction and the discussion sections to highlight this feature of our study.

Lines 80-83: Following diagnosis, TB patients may die or be lost before linking to TB care with or without having initiated TB treatment. These patients may remain unreported or return to health services for repeated testing and may eventually be included in treatment cohorts.

Lines 392-398: Previous studies based on paper-based treatment registers (9-12) were not able to track patients who returned to care after an initial test and may have over-estimated ILTFU. By using the electronic sources with a unique identifier, we were able to identify patients who returned for repeat testing and accessed treatment at any public facility in the Western Cape Province. This is a strength of the study as we were able to more accurately ascertain those patients who were classified as ILTFU and never returned to care.

2) As the primary outcome is mortality, could the authors provide some assurance of the completeness and accuracy of data linkage with the vital registration system? E.g. by referencing previous validation studies?

We have updated the methods to specify that we have verified the vital status of patients recorded in the Provincial Health Data Centre with the South African vital registration system and have clarified this in the methods and discussion.

Lines 146-148: For all TB patients who did not link to TB services, the vital status was verified in the South African population register using personal identifiers as recorded in the PHDC.

Lines 398-403: Searching for ILTFU TB patients in the population vital statistics register is a further strength, as earlier work showed that the South African vital registration system has a high level of completeness for mortality data (28). We acknowledge that the search was based on individual patient details and that the difficulties of matching using names and surnames with missing identification numbers may have under-ascertained mortality.

Minor points

1) Line 63: incidence probably better expressed per 100,000 population, or instead reword to be “an estimated 360,000 people developed active TB”.

We have updated as follows:

Lines 68-70: South Africa is one of 8 countries which jointly carry two-thirds of the global TB burden (1). An estimated 360,000 people developed active TB, the HIV prevalence was 58% in reported TB patients; and the estimated TB case fatality ratio (CFR) was 17% in 2019 (1).

2) Figure 3: could the authors put numbers at risk at each time point under the KM plots?

We have added the numbers at risk to the plots. Figure 3 is now difficult to read as a single figure with 3 panels. We have therefore produced 3 different figures (Fig 3-5) and updated the in-text references.

Reviewer #3: Reviewer Reports:

This study examined early mortality in tuberculosis patients initially lost to follow up following diagnosis in two provincial hospitals in Western Cape- South Africa. The study represents interesting findings and adds to the growing body of evidence on the true burden of TB associated mortality.

Thank you for this comment.

General Comment:

The paper includes, in its definition of ITLFU both true initial loss to follow-up (no documented TB treatment initiation within a defined time period) and early on-treatment loss to follow-up (documented TB treatment initiation but no documented continuation on TB treatment). The combination of these two definitions makes the paper difficult to read and the results difficult to follow.

The general recommendation would be for the paper to focus on true initial loss to follow-up (no documented TB treatment initiation within a defined time period) vs all those initiated on TB treatment. This eliminate the one of the significant limitations of this study which is the inability to assess reverse causality between ILTFU and early mortality among those who did initiate TB treatment within the recommended time.

Alternatively, the title could be changed to “Early mortality in patients diagnosed with tuberculosis in two provincial hospitals in Western Cape- South Africa”.

Thank you for this recommendation. We feel it is important to retain the definition as we have proposed. In the systematic review we have cited, there was no consensus on the time after diagnosis to be used for the definition of initial loss to follow up. Currently, the WHO TB reporting framework uses 1 month of treatment to define a TB treatment episode. We therefore opted to use a period of 1 month between TB diagnosis and linkage to TB services to define initial loss to follow up. The current global and South African national programmatic definition of TB requires patients to be recorded in a TB treatment register for reporting purposes. In our setting, patients diagnosed with TB in hospital who initiate TB treatment are typically given a maximum of 14 days of TB treatment upon discharge. We do not have the details of the number of doses received for each individual patient, but have programmatic reports which support this practice of providing a maximum period of 14 days. Patients who are initiated on TB treatment but do not link with the primary healthcare clinic (PHC) TB services in South Africa are not recorded in the existing TB treatment register and are therefore classified as being initial loss to follow up (ILTFU) by the TB programme. A published TB care cascade for South Africa has used this same definition to quantify initial loss to follow up. We feel that definition is appropriate, is methodologically consistent with prior work, and is programmatically relevant. 

It is important to note that: 

1. ~40% of the patients we defined as ILTFU were diagnosed with TB in hospital and were started on TB treatment in hospital 

2. ~50% of deaths among the patients we defined as being ILTFU had started their TB treatment in hospital

Furthermore, it is important to highlight, based on our data, that treatment initiation alone is insufficient to ensure that patients are linked to continued treatment and care and that they are appropriately registered, following a diagnosis and initial treatment initiation in hospital. We therefore feel that if we revise the current definition of ILTFU to reflect TB treatment initiation, ILTFU will incorrectly exclude those patients who have started an initial course of treatment in hospital and are then lost to follow up, not continuing their care, and not registered by the programme. This presents a major gap in the continuity of care for TB patients and this data is highly relevant for TB programmes. 

We have now clarified the more consistent use of the definition of ILTFU throughout the manuscript:

1. In the abstract: We aimed to evaluate mortality among TB patients who did not link to a TB treatment facility for TB treatment within 30 days of their TB diagnosis, i.e. who were “initial loss to follow-up (ILTFU)”….

2. In the introduction: We defined “initial loss to follow-up (ILTFU)” as TB patients who did not link to a TB treatment facility (primary health care (PHC) facility or TB hospital) for TB treatment within 30 days of their TB diagnosis having been made.

3. In the Methods section, referencing table 1: TB patients who did not link with a TB treatment facility (PHC facility or TB hospital) in the province within 30 days of their TB diagnosis for TB treatment, regardless of whether they had initiated their TB treatment in hospital. Linkage to a TB treatment facility included attendance for TB treatment at a PHC facility or admission to a TB hospital following the TB diagnosis. TB patients defined as being ILTFU included those diagnosed at a TB treatment facility who neither linked to care at a different TB treatment facility nor returned to care at the facility where they were diagnosed and patients who had died before their linkage to TB care.

Methods:

The study includes both patients with drug susceptible TB (DSTB) and those with multidrug resistant TB (MDR TB) are mutually exclusive cohorts. DSB and MDRTB tend to have different paths to treatment initiation/registration. As seen in your data (Table 3), initial loss to follow-up was higher among MDR TB patients diagnosed at PHC than those diagnosed in hospital. In addition, MDR TB treatment is longer so a follow-up period of 7 months will not capture all mortality during TB treatment (the definition used in this study). It is therefore not appropriate to combine both groups of patients in the same analysis. Recommendation would be to include only one group of patients for this analysis (DS TB)

Thank you for this valuable comment and agree with the reviewer. We have now excluded patients with confirmed DR-TB from this analysis; this has reduced the size of the overall cohort by 5%. We have updated the methods and the results section to reflect this change. We have noted in the discussion that future work on DR TB is needed in this regard.

As ILFU more commonly means patient who are not initiated on TB treatment within a set period of time, it seems more appropriate to include treatment initiation or the lack thereof in your adjusted model. If there is co-linearity between treatment initiation and healthcare level (and there is), it might be the better choice to include treatment initiation and leave out healthcare level. If this is not possible, please explain why.

Thank you for this comment. We have revised our final model accordingly.

ILTFU, treatment initiation and healthcare level were all included as independent variables in the univariate analyses. Based on the co-linearity (which the reviewer has correctly noted) we did not include these 3 variables as independent variables in the final model. 

We rather created the composite variable “Level of care of diagnosis, treatment status, and ILTFU”, which is now included in the final model and provides the adjusted estimates of odds of mortality based on the combination of ILTFU, treatment initiation and healthcare level at which diagnosis was made.

We believe that this approach is more appropriate than the inclusion of treatment initiation independently, as it combines the complexity of ILTFU, the level of care and treatment initiation in a combined variable. As we have also noted under previous comments, treatment initiation status alone is not sufficient to ensure that patients are linked to continued treatment and care and appropriate registration, following a diagnosis and initial treatment initiation in hospital.

We have updated the results section, 

lines: 207-210

…. and 21% (221/1,055) of those diagnosed in hospital who started TB treatment died (Table 4). Among those diagnosed in hospital who started TB treatment in hospital but never linked to ongoing care 27% (209/786) died.

lines: 230-244

Compared to TB patients diagnosed at PHC facilities who started TB treatment and were linked to TB services, TB patients diagnosed in hospital had an increased odds of mortality. TB patients diagnosed in hospital who started their TB treatment and were linked to ongoing TB services had an aOR: 3.29 (95%CI: 2.62-4.13), while those who started their TB treatment and were ILTFU had an aOR: 9.53 (95%CI: 7.56-12.02), and those who did not start their TB treatment and were ILTFU had an aOR: 28.66 (95%CI: 21.53-38.15) (Table 4). ILTFU following a TB diagnosis in PHC facilities also increased the odds of mortality, and this was higher in those who did not start their TB treatment, aOR: 3.48 (95%CI:2.45-4.94) as opposed to those who had been started on TB treatment, aOR: 1.47 (95%CI: 0.9-2.41) (Table 4).

We have also noted the limitations of the use of the definitions and our additional analyses in the limitations section:

Lines 405-411

We acknowledge that we did not distinguish the exact time to treatment initiation, or the place of TB treatment initiation and that the group of TB patients who were diagnosed in hospital and had started treatment and were ILTFU included both a group of patients who never linked to care as well as patients who did link to TB care after 30 days. Additional work to evaluate the level of care at which TB treatment is initiated, the time to treatment initiation, the duration of the initial treatment provided and the time taken to link to TB treatment services after initial hospital admission is needed.

Result:

There is no mention on the effect of other co-morbidities (apart from HIV) e.g. diabetes on mortality following TB diagnosis in the univariate or bivariate analysis.

Thank you for this important comment. We have shown the proportion of individuals with diabetes in table 2 and 3 and have now also included diabetes in the UV and MV model. We have noted the increased odds of mortality in individuals with diabetes, and that it was not maintained in the final model. We have now stated the following, in the discussion section:

Line 370-375: The odds of mortality increased with a diagnosis of diabetes in univariate analyses, but this effect was not maintained in the multivariable model. This may relate to the higher proportion of TB patients with diabetes diagnosed at hospital level (8.7%) compared to the proportion of TB patients with diabetes diagnosed at PHC facilities (5.3%). Further work examining the effect of diabetes and other co-morbidities on ILTFU and mortality is needed.

Tables are crowded. Legibility could be improved by deleting the column marked total (farthest to the left) in Tables 2 and 3

Thank you for the suggestion. We have deleted the total columns in table 2 and 3. We have also removed the % symbol to improve the legibility.

Figure 3 is illegible. 

We have updated figure 3 to include at-risk numbers and have split figure 3 into 3 separate figures (Fig 3-5)

Discussion

In a number of countries, patients are registered in the TB treatment register on initiation of TB treatment. This might be a recommendation this study makes so that early on-treatment mortality is recorded in the national registry and used to guide TB case-finding interventions

Thank you, we have included this in the discussion, as follows: 

Discussion Line 355-362: In our setting, the registration of TB patients in hospital at the time of diagnosis can take place through the integrated PHDC. However, in other settings, the registration of TB patients in-hospital at the time of diagnosis is an important intervention that is required. The inclusion of these patients in routine programmatic reporting is essential to ensure that early on-treatment mortality is recorded and to guide TB case-finding interventions to ensure earlier care and prevent mortality. We have shown that following this registration, TB treatment initiation and linkage to TB services for continued treatment and care remain critical steps to further reduce TB mortality.

Conclusion Line 430-432: The registration of TB patients in-hospital at the time of their diagnosis is an important intervention to support linkage to TB services to reduce early mortality and ensure that when occurring, early on-treatment mortality is recorded.

A number of studies -including prevalence surveys- show that a) patients seek care for TB symptoms multiple times before being diagnosed with TB and that b) a high proportion of patients who present to primary care facilities with signs and symptoms of TB do not get tested with. It is therefore likely that the presentation to a hospital with advanced TB disease partly represents a failure of the health system. Kindly include this in your discussion.

Thank you. We agree with this important comment. We have added this to the discussion, lines 325-327:

High numbers of patients presenting to hospitals with advanced TB disease might reflect barriers in accessing primary health care at an earlier stage of TB disease. This may highlight opportunities for earlier TB case detection, before hospital admission, and could include active case finding

Line 204: the more appropriate representation of the results would be mortality was only ascertained in 49 out of 100 patients who were not initiated on TB treatment within two weeks of diagnosis or 49% of patients not initiated on TB treatment within two weeks of diagnosis.

We agree and have updated this as follows:

Lines 303-305: In Uganda, 22.4% of pre-treatment loss to follow up TB patients died, but mortality was only ascertained in 49 out of 100 patients with a positive Xpert results who were not initiated on TB treatment within two weeks of diagnosis.

Please provide references this statement in Lines 242 and 243 The importance of rapid treatment initiation is unequivocal for the patient and reducing transmission. There are many studies particularly among HIV+ patients that compared early to later TB treatment initiation.

We have updated this statement and have included the following references:

1. Asres A, Jerene D, Deressa W. Delays to treatment initiation is associated with tuberculosis treatment outcomes among patients on directly observed treatment short course in Southwest Ethiopia: a follow-up study. BMC Pulm Med 2018; 18(1): 64.

2. Gebreegziabher SB, Bjune GA, Yimer SA. Total Delay Is Associated with Unfavorable Treatment Outcome among Pulmonary Tuberculosis Patients in West Gojjam Zone, Northwest Ethiopia: A Prospective Cohort Study. PLoS One 2016; 11(7): e0159579.

3. Virenfeldt J, Rudolf F, Camara C, et al. Treatment delay affects clinical severity of tuberculosis: a longitudinal cohort study. BMJ Open 2014; 4(6): e004818.

4. Dowdy DW, Davis JL, den Boon S, Walter ND, Katamba A, Cattamanchi A. Population-level impact of same-day microscopy and Xpert MTB/RIF for tuberculosis diagnosis in Africa. PLoS One 2013; 8(8): e70485.

5. Cheng S, Chen W, Yang Y, et al. Effect of Diagnostic and Treatment Delay on the Risk of Tuberculosis Transmission in Shenzhen, China: An Observational Cohort Study, 1993-2010. PLoS One 2013; 8(6): e67516.

---

## [Decision Letter · Decision Letter 1]

10 May 2021

Early mortality in tuberculosis patients initially lost to follow up following diagnosis in provincial hospitals and primary health care facilities in Western Cape, South Africa

PONE-D-20-40984R1

Dear Dr. Osman,

We’re pleased to inform you that your manuscript has been judged scientifically suitable for publication and will be formally accepted for publication once it meets all outstanding technical requirements.

Kind regards,

Richard John Lessells, BSc, MBChB, MRCP, DTM&H, DipHIVMed, PhD

Academic Editor

PLOS ONE

Additional Editor Comments (optional):

Reviewers' comments:

Reviewer's Responses to Questions

**Comments to the Author**

1. If the authors have adequately addressed your comments raised in a previous round of review and you feel that this manuscript is now acceptable for publication, you may indicate that here to bypass the “Comments to the Author” section, enter your conflict of interest statement in the “Confidential to Editor” section, and submit your "Accept" recommendation.

Reviewer #1: All comments have been addressed

Reviewer #2: All comments have been addressed

Reviewer #3: All comments have been addressed

2. Is the manuscript technically sound, and do the data support the conclusions?

Reviewer #1: (No Response)

Reviewer #2: Yes

Reviewer #3: (No Response)

3. Has the statistical analysis been performed appropriately and rigorously? 

Reviewer #1: (No Response)

Reviewer #2: Yes

Reviewer #3: (No Response)

4. Have the authors made all data underlying the findings in their manuscript fully available?

Reviewer #1: (No Response)

Reviewer #2: Yes

Reviewer #3: (No Response)

5. Is the manuscript presented in an intelligible fashion and written in standard English?

Reviewer #1: (No Response)

Reviewer #2: Yes

Reviewer #3: (No Response)

6. Review Comments to the Author

Reviewer #1: (No Response)

Reviewer #2: (No Response)

Reviewer #3: (No Response)

7. PLOS authors have the option to publish the peer review history of their article (what does this mean?). If published, this will include your full peer review and any attached files.

Reviewer #1: No

Reviewer #2: **Yes: **Peter MacPherson

Reviewer #3: No

---

## [Editor Report · Acceptance letter]

14 May 2021

PONE-D-20-40984R1 

Early mortality in tuberculosis patients initially lost to follow up following diagnosis in provincial hospitals and primary health care facilities in Western Cape, South Africa 

Dear Dr. Osman:

I'm pleased to inform you that your manuscript has been deemed suitable for publication in PLOS ONE. Congratulations! Your manuscript is now with our production department. 

Kind regards, 

on behalf of

Dr. Richard John Lessells 

Academic Editor

PLOS ONE